# Natural Pectin-Based Edible Composite Coatings with Antifungal Properties to Control Green Mold and Reduce Losses of ‘Valencia’ Oranges

**DOI:** 10.3390/foods11081083

**Published:** 2022-04-09

**Authors:** María Victoria Alvarez, Lluís Palou, Verònica Taberner, Asunción Fernández-Catalán, Maricruz Argente-Sanchis, Eleni Pitta, María Bernardita Pérez-Gago

**Affiliations:** 1Grupo de Investigación en Ingeniería en Alimentos, Departamento de Ingeniería Química y en Alimentos, Facultad de Ingeniería, Universidad Nacional de Mar del Plata, Consejo Nacional de Investigaciones Científicas y Técnicas (CONICET), Mar del Plata 7600, Argentina; mvalvarez@fi.mdp.edu.ar; 2Centre de Tecnologia Postcollita (CTP), Institut Valencià d’Investigacions Agràries (IVIA), 46113 València, Spain; taberner_ver@gva.es (V.T.); fernandez_asucat@externos.gva.es (A.F.-C.); argente_mcrsan@gva.es (M.A.-S.); elenipitta15@gmail.com (E.P.); perez_mbe@gva.es (M.B.P.-G.); 3School of Agriculture, Faculty of Agriculture, Forestry and Natural Environment, Aristotle University of Thessaloniki, 54124 Thessaloniki, Greece

**Keywords:** *Penicillium digitatum*, citrus, essential oils, natural extracts, edible coatings, postharvest decay

## Abstract

Novel pectin-based, antifungal, edible coatings (ECs) were formulated by the addition of natural extracts or essential oils (EOs), and their ability to control green mold (GM), caused by *Penicillium digitatum*, and preserve postharvest quality of ‘Valencia’ oranges was evaluated. *Satureja montana*, *Cinnamomum zeylanicum* (CN), *Commiphora myrrha* (MY) EOs, eugenol (EU), geraniol (GE), vanillin, and propolis extract were selected as the most effective antifungal agents against *P. digitatum* in in vitro assays. Pectin-beeswax edible coatings amended with these antifungals were applied to artificially inoculated oranges to evaluate GM control. ECs containing GE (2 g/kg), EU (4 and 8 g/kg), and MY EO (15 g/kg) reduced disease incidence by up to 58% after 8 days of incubation at 20 °C, while CN (8 g/kg) effectively reduced disease severity. Moreover, ECs formulated with EU (8 g/kg) and GE (2 g/kg) were the most effective on artificially inoculated cold-stored oranges, with GM incidence reductions of 56 and 48% after 4 weeks at 5 °C. Furthermore, ECs containing EU and MY reduced weight loss and maintained sensory and physicochemical quality after 8 weeks at 5 °C followed by 7 days at 20 °C. Overall, ECs with EU were the most promising and could be a good natural, safe, and eco-friendly commercial treatment for preserving orange postharvest quality.

## 1. Introduction

Citrus fruits are among the most relevant horticultural crops in the world. Significant economic losses occur after harvest due to fruit quality deterioration, mainly caused by fungal infections and weight loss [1]. The fungus *Penicillium digitatum* (Pers.:Fr.) Sacc., causing citrus green mold (GM), is the most important postharvest pathogen of citrus fruits worldwide and, particularly, in Mediterranean climate regions [2]. It is a strict wound pathogen that infects the fruit through rind microwounds caused by insects, branches, or inadequate handling during harvest and postharvest operations [2,3]. Conventional synthetic waxes, often amended with chemical fungicides, have been traditionally used by the citrus industry to reduce postharvest decay and extend fruit shelf life [4,5]. However, the continued application of these treatments has resulted in serious problems, such as health and environmental issues associated with conventional commercial waxes containing ammoniacal compounds or synthetic fungicides that generate chemical residues and lead to the development of resistant fungal strains [5,6,7]. In this sense, food-safety regulations from many countries are increasingly restricting the use of synthetic fungicides and ingredients used in commercial wax coatings [3,4]. Likewise, traditional and new organic citrus markets are increasingly demanding products with lower or zero levels of pesticides in order to satisfy the safety demands from consumers. Therefore, new research prioritizing the development of safer and eco-friendly alternative strategies to control postharvest diseases and reduce losses is needed. Such alternatives include natural edible coatings (ECs) with antifungal properties [4].

Major components of composite ECs include proteins or polysaccharides and lipids [4]. When properly combined, these ingredients form a thin layer around the fruit, creating a semi-permeable barrier against gases and water vapor that contributes to maintain the weight, firmness, and other quality attributes of coated fruits during storage [5,8]. The antifungal activity of ECs is mainly accomplished through the incorporation to the formulations of non-polluting antifungal ingredients, such as essential oils (EOs), plant extracts, food additives, low toxicity compounds classified as generally recognized as safe (GRAS), and microbial antagonists as biocontrol agents [1,3,9,10]. Among them, many EOs and natural extracts have proven antifungal activity against important postharvest fungal pathogens. The most studied EOs for fruit preservation have been clove, oregano, thyme, and cinnamon [4,11,12,13]. However, their application as stand-alone treatments as volatiles or in aqueous solutions have shown limitations, such as lower antifungal activity in in vivo than in in vitro studies and induction of phytotoxic effects or negative sensory properties to treated fruits [5,9]. The incorporation of natural extracts or EOs as ingredients of the formulation of composite ECs might be an effective approach to exploit their antifungal potential and solve some of these application problems. Thus, a gradual migration of the bioactive compounds from the coating matrix to the fruit surface could allow to maintain their activity throughout postharvest storage [14,15].

Several studies have reported the antifungal potential of natural extracts or EOs to control postharvest diseases of citrus fruits. Most of them are focused on the use against green or blue molds of aqueous solutions of extracts from sources such as propolis, seaweed, garlic, and clove [12,16,17,18,19] or the application of conventional commercial waxes based on carnauba or shellac, amended with volatile compounds such as carvacrol, thymol, citral, and cinnamaldehyde, among other EOs [6,11,15,20,21]. However, only a few studies reported the application of biopolymer-based ECs functionalized by the incorporation of this type of natural antifungal compounds for controlling citrus decay and preserving quality [1,14,22,23]. Most of these studies used chitosan as the base polysaccharide, whereas many other biopolymer matrices, such as citrus pectin, remain unexplored. In addition, among plant-derived extracts and pure compounds, a large number have not been studied as potential antifungal agents to control citrus postharvest diseases. In this work, several natural plant extracts and EOs were studied as potential ingredients of antifungal pectin (PEC)-based ECs to control GM and maintain quality of ‘Valencia’ oranges. Specific objectives were to: (a) evaluate the in vitro antifungal activity of natural agents against *P. digitatum*; (b) determine the curative activity of PEC-beeswax ECs formulated with selected natural antifungal agents against GM on artificially inoculated oranges incubated at 20 °C; (c) evaluate the effectiveness of selected antifungal ECs to control GM on inoculated oranges during long-term storage at 5 °C; and (d) assess the impact of these treatments on postharvest quality of cold-stored fruit, followed by a 7-day shelf-life period at 20 °C, in terms of weight loss, firmness, juice quality, volatiles content (ethanol and acetaldehyde), and sensory properties.

## 2. Materials and Methods

### 2.1. Commercial Essential Oils (EOs) and Natural Extracts

Cinnamon EO (CN, *Cinnamomum zeylanicum*), lemongrass EO (LG, *Cymbopogon citratus*), and natural pure compounds, such as eugenol (EU), geraniol (GE), and vanillin (VA) were purchased to Sigma-Aldrich (St. Louis, MO, USA). *Satureja montana* EO (SM) was provided by Essential’ Arôms (Lleida, Catalonia, Spain) and myrrh EO (MY, *Commiphora myrrha*) by Essenciales (Barcelona, Catalonia, Spain). Dry extracts of green tea (GT; *Camellia sinensis* L.) and propolis (PRO) were obtained from Guinama (Valencia, Spain).

### 2.2. Fugal Pathogen and Inoculum Preparation

*Penicillium digitatum* NAV-7 was isolated from decayed oranges from local packinghouses in the Valencia region (Spain), identified, and maintained in the IVIA culture collection of postharvest pathogens. This strain was deposited in the Spanish Type Culture Collection (CECT, University of Valencia, Valencia, Spain) with the accession number CECT 21008. Before the experiments, the fungus was incubated on potato dextrose agar (PDA) (Scharlab S.L., Barcelona, Spain) in Petri dishes for 7–14 days at 25 °C. High-density conidial suspensions were prepared following the methodology described by Soto-Muñoz et al. [2] and then diluted with sterile water to achieve the desired inoculum density for each in vitro and in vivo assay.

### 2.3. In Vitro Antifungal Activity of Natural Agents against P. digitatum

According to the chemical nature of the antifungal compounds, two different methods were used for antifungal activity assessment. The volatile exposure method was used to test the activity of commercial EOs and pure volatile compounds according to Plaza et al. [11] SM, CN, LG, EU, and GE were applied at doses of 10, 20, and 40 µL by soaking 55 mm sterile filter paper discs placed in the lid of 90 mm diameter PDA Petri dishes inoculated with *P. digitatum*. Petri dishes were sealed with Parafilm and incubated upside-down. Sterile distilled water was used for control dishes. The agar dilution method was used for GT, PRO, VA, and MY following the methodology described by Martínez-Blay et al. [24] with some modifications. GT was previously dissolved in sterile distilled water, PRO in ethanol–water 80:20, and VA and MY in pure dimethyl sulfoxide (DMSO). Appropriate amounts of these mixtures were added to PDA to achieve final concentrations of 5, 10, and 20 g/kg GT; 5 and 10 g/kg PRO; 0.62, 1.25, and 2.5 g/kg VA; and 1.25, 2.5, and 5 g/kg MY. Control dishes used were PDA, PDA amended with ethanol–water 80:20 (50 g/kg), and PDA amended with DMSO (5 g/kg). For both methods, each dish was inoculated by adding 20 µL of a 10^9^ spores/L suspension of *P. digitatum* in the center of the agar. The plates were incubated for 7 days at 25 °C in the dark. Radial mycelial growth was determined in each plate by calculating the mean of two perpendicular fungal colony diameter measurements. Five replicates, each one corresponding with one 90 mm Petri dish, were used for each antifungal agent and concentration. Results were expressed as the percentage of mycelial growth inhibition according to the formula: [(dc − dt)/dc] × 100, where dc = average diameter of the fungal colony on the corresponding control plates, and dt = average diameter of the fungal colony on Petri dishes subjected to each treatment.

### 2.4. Preparation of Antifungal Edible Coatings

Pectin (PEC)–lipid composite ECs were prepared by combining the biopolymer (citrus pectin—Ceampectin RS 4710, DE 70–75%; CEAMSA, Pontevedra, Spain) with beeswax (BW; Guinama, Valencia, Spain) as lipidic component suspended in water. Glycerol was used as plasticizer and a mixture of oleic acid and palmitic acid at a ratio 1:1 (Panreac Química SA, Barcelona, Spain) as emulsifiers. All the formulations contained PEC at 20 g/kg and BW at 7 g/kg with ratios of PEC–glycerol and BW–emulsifiers of 2:1 and 5:1, respectively. To prepare EC emulsions, BW, glycerol, emulsifiers, and water were added to an aqueous solution of pectin previously prepared, and the mixture was heated to 92 °C to melt the lipid. Samples were homogenized using a high-shear probe mixer (Ultra-Turrax IKA^®^ model T25, IKA-Werke, Staufen, Germany) for 1 min at 12,000 rpm and 3 min at 22,000 rpm [8]. EOs and natural extracts, selected from the previous in vitro screening assays, were incorporated at non-phytotoxic concentrations defined in preliminary coating-application tests (data not shown). Thus, SM, CN, EU, and GE were added to the PEC-based ECs at 2, 4, and 8 g/kg; VA at 2.5, 5, and 10 g/kg; PRO at 5, 10, and 20 g/kg; and MY at 15 g/kg. These mixtures were homogenized for 2 min at 16,000 rpm at room temperature. All the prepared antifungal ECs formed stable emulsions, with viscosity values in the range of 50–162 mPa·s and pH values of 3.11–3.38.

### 2.5. Fruit

‘Valencia’ oranges (*Citrus sinensis* (L.) Osbeck) were harvested in commercial orchards in Valencia (Spain) and stored up to 1 week at 5 °C and 90% relative humidity (RH) before use. Fruits were selected for uniformity of size and shape and used in the experiments before any postharvest treatment was applied. Selected fruits were surface disinfected by immersion in a 5 g/L sodium hypochlorite solution for 4 min, rinsed with tap water, and allowed to air dry at room temperature.

### 2.6. In Vivo Green Mold Control by Antifungal Coatings

Oranges were artificially inoculated with *P. digitatum* (inoculum concentration of 10^8^ conidia/L) by immersing a stainless steel rod with a probe tip 1 mm wide and 2 mm in length into the conidial suspension and wounding each fruit once on the equator. Inoculated fruit were incubated for 24 h at 20 °C and 90% RH. After this period, coatings were applied by adding 0.4 mL of the coating material onto each fruit and rubbing with gloved hands to simulate the application of industrial coating machinery in packing lines in citrus packinghouses. Control treatment included inoculated but uncoated oranges. In addition, a PEC-based coating without the addition of antifungal agents (PEC) was also evaluated. Fruits were allowed to air dry at room temperature, placed on plastic trays, and incubated for up to 12 days at 20 °C and 90% RH. During this period, disease incidence (%) was determined by counting the number of decayed fruits (fruits showing visible disease symptoms) and disease severity as the diameter of the lesion (mm). Four replicates of five fruits each were used for each coating treatment. Results obtained at day 8 are presented.

### 2.7. Effect of Selected Antifungal Coatings on Decay and Quality of Cold-Stored Oranges

PEC-BW coatings containing CN at 8 g/kg (PEC-CN), EU at 8 g/kg (PEC-EU), GE at 2 g/kg (PEC-GE), and MY at 15 g/kg (PEC-MY) were selected among those previously tested to be used in these cold storage evaluations. In addition, an uncoated control and a PEC treatment were included. Fruits were prepared as described above and randomly distributed into two lots: the first for pathology tests involving fungal inoculation and the second for quality evaluations of non-inoculated fruit.

#### 2.7.1. Green Mold Control on Cold-Stored Fruit

Oranges were artificially inoculated with *P. digitatum* and coated 24 h later as described above. Four replicates of ten fruits each were used for each coating treatment. GM incidence (%) and severity (mm) were assessed after 2, 4, and 6 weeks of storage at 5 °C and 90% RH following the methodology previously described (Section 2.6).

#### 2.7.2. Quality of Cold-Stored Fruit

Lots of 70 oranges per treatment were manually coated with selected antifungal ECs as described previously. Control fruits were dipped for 15 s in tap water at 20 °C. The following quality attributes were assessed in oranges at harvest and after 4 and 8 weeks of cold storage at 5 °C and 90% RH plus 1 week of shelf life at 20 °C (35 fruits per each storage period), following the methodology described by Valencia-Chamorro et al. [25]:

*Fruit weight loss*. Twenty oranges per treatment were individually weighed at the beginning and the end of each storage period with a calibrated analytical balance. Results are expressed as the percentage of initial weight loss.

*Firmness*. Firmness of 10 oranges per treatment was determined after each storage period using an Instron Universal testing machine (Model 3343, Instron Corp., Canton, MA, USA). Each fruit was compressed between two flat surfaces closing together at a rate of 5 mm/min. The machine measured the deformation (mm) after application of a load of 10 N to the equatorial region of the fruit. Results are expressed as percentage of rind deformation related to initial fruit diameter.

*Internal quality*. Titratable acidity (TA, g/L of citric acid) and soluble solids content (SSC, °Brix) were determined in 5 mL juice samples from squeezed oranges (three replicates of five fruit each per treatment and storage time). TA was determined using an automatic titrator (Titrator T50, Mettler Toledo, Switzerland), and a digital refractometer (model ATC-1, Atago^®^ Co., LTD, Tokyo, Japan) was employed for SSC measurements.

*Ethanol and acetaldehyde content*. Ethanol and acetaldehyde concentrations were determined from the headspace of juice samples using a gas chromatograph (GC) (Thermo Trace, Thermo Fisher Scientific, Waltham, MA, USA) equipped with a flame ionization detector (FID) and 1.2 m × 0.32 cm (i.d.) Poropack QS 80/100 column. The injector was set at 175 °C, the column at 150 °C, the detector at 200 °C, and the carrier gas at 28 mL/min. Three juice replicates from five oranges per treatment each were prepared. Five milliliters of juice were transferred to 10 mL vials with crimp-top caps and TFE/silicone septum seals. Samples were frozen and stored at −18 °C until analyses. At the time of analysis, a 1 mL sample of the headspace was withdrawn from vials, previously equilibrated in a water bath at 20 °C for 1 h, followed by 10 min at 40 °C to reach equilibrium in the headspace, and then injected into the GC [25]. Ethanol and acetaldehyde were identified by comparison of retention times with standards and results are expressed as mg/L.

*Sensorial evaluation*. Ten semi-trained panelists evaluated sensory quality of coated oranges. Overall taste (1–9 scale, where 1 to 3 represented a range of non-acceptable quality with the presence of off-flavor, 4 to 6 represented a range of acceptable quality, and 7 to 9 represented a range of excellent quality) and the presence of off-flavors (1–5 scale, from 1 = absence to 5 = very pronounced) were evaluated in orange segments. Panelists also evaluated external appearance of coated and uncoated oranges (1–3 scale: 1 = bad, 2 = acceptable, and 3 = good) and visually ranked the treatments from the highest to the lowest gloss.

### 2.8. Statistical Analysis

A completely randomized design was used for all experiments. Specific differences between means were determined by Fisher’s protected least significant difference test (LSD, *p* < 0.05) applied after an analysis of variance (ANOVA). Disease incidence data were arcsine transformed. Friedman test (*p* < 0.05) was used for ranking fruit gloss. In all cases, data shown are means ± standard errors. Analyses were performed with the software Statgraphics Centurion XVII (Statgraphics Technologies Inc., The Plains, VA, USA).

## 3. Results

### 3.1. In Vitro Antifungal Activity

Table 1 shows the effect of the different antifungal agents on the in vitro mycelial growth of *P. digitatum* in both volatile exposure and agar dilution tests. SM and EU were the most effective agents applied as volatiles, with a radial growth inhibition of 90–100% at a dose of 10 µL, followed by CN and GE, which caused the same inhibitory effect at a dose of 20 µL. LG showed low effectiveness against *P. digitatum* at all tested concentrations. MY EO showed no activity when tested in the vapor phase (data not shown), so it was evaluated by the agar dilution method, adding this EO as aqueous solution into the culture medium. Among the natural antifungal agents tested by the agar dilution method, VA and PRO induced 100% growth inhibition at 1.25 and 5 g/kg, respectively, while MY EO caused a moderate effect that did not depend on the concentration. GT was effective only when applied at the highest concentration (20 g/kg). Thus, SM, CN, EU, GE, VA, PRO, and MY were the natural agents selected to be tested in the subsequent in vivo assays.

### 3.2. In Vivo Green Mold Control

Figure 1 shows the development of GM after 8 days of incubation at 20 °C on ‘Valencia’ oranges artificially inoculated with *P. digitatum.* Disease incidence on both uncoated oranges and those coated with PEC-BW (without antifungal agent) was very high (90–95%), indicating that ECs formulated without agents were not antifungal. ECs containing GE (2 g/kg), EU (4 and 8 g/kg), or MY (15 g/kg) significantly (*p* < 0.05) reduced GM incidence compared to uncoated and PEC-coated fruit, with reductions of up to 58%. ECs formulated with SM, CN, VA, or PRO did not reduce GM incidence. ECs containing CN (8 g/kg), EU (4 g/kg), and GE, at all tested concentrations, reduced GM severity by 44 to 55% compared to uncoated controls, while it was not significantly reduced by the rest of treatments. From these results, ECs containing CN (8 g/kg), EU (8 g/kg), GE (2 g/kg), or MY (15 g/kg) were selected to be tested under commercial cold-storage conditions.

### 3.3. Impact of Selected Coatings on Decay and Quality of Cold-Stored Oranges

#### 3.3.1. Green Mold Control

Figure 2 shows the incidence and severity of GM on coated and uncoated oranges stored at 5 °C for up to 6 weeks. After 2 weeks of cold storage, all the antifungal ECs reduced disease incidence compared to control samples. After 4 weeks, ECs formulated with EU and GE significantly reduced GM incidence by 56 and 48%, respectively, while reductions by CN and MY were not significant (*p* > 0.05). At the end of the storage period, disease incidence increased for all treatments, and no significant differences were observed between coated and uncoated control oranges. Nevertheless, EU and GE coatings induced lower disease incidence than the coating without antifungals. Regarding GM severity, ECs containing EU were the most effective in reducing it and significantly (*p* < 0.05) inhibited fungal growth during the entire cold-storage period, with severe reductions of 70 and 46% compared to the control after 4 and 6 weeks, respectively. CN and GE coatings were less effective, with significant reductions of 47–54% after 4 weeks.

#### 3.3.2. Physicochemical and Sensory Quality

Weight loss of ‘Valencia’ oranges ranged from 2.9 to 3.8% after 8 weeks of cold storage followed by 7 days of shelf life at 20 °C (Figure 3). The PEC coating without antifungal agents did not reduce weight loss compared to uncoated control fruit (*p* > 0.05), whereas, in contrast, ECs containing EU, GE, or MY significantly reduced weight loss. In general, ECs with EU and MY were the most effective and reduced weight loss by 19 and 22%, respectively, at the end of the storage period.

Physicochemical and sensory quality of fruits at harvest and after 4 and 8 weeks of cold storage followed by a shelf-life period of 1 week at 20 °C is shown in Table 2. Fruit firmness ranged from 2.4 to 4.0% rind deformation. In general, firmness decreased (higher rind deformation) during storage, but the ECs did not affect this quality attribute, with the exception of the PEC-CN coating that induced lower rind deformation compared to control at the end of storage. TA was not significantly affected by coating application after 4 weeks, and only PEC-coated oranges showed slightly higher TA (8.90 mg/L) compared to uncoated samples (7.17 mg/L) at the end of storage. No significant differences in SSC were observed among oranges subjected to the different treatments after the entire storage period, with values from 10.1 to 11.0 °Brix. 

The concentration of ethanol and acetaldehyde in the juice increased with storage time from 153.6 and 5.0 mg/L at harvest to 434.7–1148.9 and 10.5–20.5 mg/L at the end of storage, respectively. The application of the PEC-based coatings significantly increased the content of both volatiles compared to uncoated samples. At the end of the experiment, no significant differences in acetaldehyde content were found among coated oranges, whereas the PEC-GE coating induced lower ethanol content in the juice than the rest of the ECs.

Sensory properties of cold-stored oranges were not negatively affected by the application of antifungal ECs. Overall flavor decreased from a score of 7.3 (good quality) at harvest to 4.4 (acceptable) at the end of storage, whereas off-flavor scores increased from 1.1 (absence) at harvest to 1.8 (very slight) after the entire storage period. Similarly, visual quality slightly decreased from 3.0 (good) to 2.4–2.6 (fair-good) at the end of the experiment. Significant differences in fruit gloss were only observed after 4 weeks at 5 °C followed by 1 week at 20 °C (Table 3). In general, control and PEC-coated (without antifungal agent) samples were ranked with the lowest gloss, and only oranges coated with the PEC-EU coating showed significantly higher gloss than fruits subjected to the rest of treatments after 4 weeks of cold storage plus the shelf-life period.

## 4. Discussion

In the current commercial scenario, alternative strategies to conventional fungicides are needed to control citrus postharvest diseases caused by *Penicillium* spp. [3]. To our knowledge, this is the first research work in which PEC-based ECs formulated with natural EOs or plant extracts are applied to citrus fruit for the control of GM caused by *P. digitatum*. Likewise, previous references to the use of conventional commercial waxes and natural ECs amended with EOs to preserve quality of cold-stored oranges are very scarce [15,23].

EOs contain combinations of volatile secondary metabolites that are active in vapor phase and can act directly against phytopathogenic fungi. In addition, some of the key active compounds of EOs can also be used as food preservatives [9]. Among the volatile compounds tested here against *P. digitatum* in the vapor phase, EO from SM showed the highest effectiveness. Similarly, the same commercial SM EO showed 100% mycelial growth inhibition against *Alternaria alternata* and other postharvest fungi when added to PDA plates at 300 µg/mL [26]. The antifungal properties of this EO were attributed to those of its major components, which included carvacrol (24%), γ-terpinene (15.9%), and p-cymene (14.2%). In the case of CN, a dose of 20 µL in the vapor phase was needed to completely inhibit the mycelial growth of *P. digitatum*. This dose was higher than that reported by Plaza et al. [11], who showed 100% inhibition of *P. digitatum* by using 5 µL of CN, and this may be explained by differences in the composition of the EOs used. The antifungal activity of CN is mainly attributed to the presence of eugenol and cinnamaldehyde [27]. Thus, in vitro studies by Combrink et al. [27] showed 100% inhibition of *P. digitatum* with CN (81% eugenol content) applied by direct contact in PDA plates at concentrations of 2–3 g/L, while a concentration of 0.5 g/L of pure EU was required for complete growth inhibition. This is also confirmed by our results, as the lowest EU dose tested (10 µL) reduced mycelial growth by more than 90%. In our study, GE at 20 µL also inhibited *P. digitatum* by 90%, which is a very good result. In a study by Plaza et al. [11], higher antifungal activity of different EOs against *Penicillium* spp. was found when they were tested in the vapor phase than when they were directly incorporated into PDA media, which was attributed to the high volatility of terpenes and phenols and their stronger and more prolonged activity in the volatile test. Hence, EOs and pure components were evaluated in our work using the volatile exposure method. Moreover, Prakash et al. [10] reported that MY EO completely inhibited the growth of nine molds, including *Penicillium italicum*, when it was added to PDA at 2.5–3.5 mL/L. However, no data were available for *P. digitatum*. In our work, MY EO was also tested using the agar method, showing a moderate inhibitory effect against GM at 1.25–5 g/kg. The commercial MY EO used here contained curzerene (40.7%), lindestrene (25.6%), and furanoeudesma-1.3-diene (8.1%) as main components (data provided by the supplier). A proposed mechanism of action of EOs against fungi involves the disruption of the cell membrane integrity and membrane permeability. The lipophilic character of EOs facilitates their infiltration into the membrane structure with several negative consequences, such as increase of membrane fluidity and permeability, disturbance of membrane-bound proteins, disruption of ion transport processes, and leakage of intercellular contents [14].

In the case of many extracts derived from plant sources, the antifungal activity has been attributed to the presence of polyphenols including phenolic acids, flavonoids, and their derivatives [3]. Thus, several studies have reported that the antimicrobial properties of GT leaf extracts are due to the high content of condensed tannins, mainly epigallocatechin gallate [28]. A commercial GT extract showed moderate effectiveness against *P. digitatum* (65% in vitro growth inhibition) when applied at 2 g/L, as reported by Romani et al. [29]. Numerous studies also reported the antifungal activity of PRO extracts [30,31]. The PRO biological activity is related to the high content of flavonoids, which can vary according to its geographical origin [31]. In agreement with our results, an Argentinian PRO ethanolic extract was found to be highly effective against *P. digitatum*, with 100% growth inhibition in PDA amended with 0.5 g/L [30]. VA is a natural phenolic aldehyde with a well-demonstrated antimicrobial activity. Thus, for example, VA has been reported as an effective inhibitor of the growth of the fungus *Botrytis cinerea* at concentrations from 0.62 g/L in in vitro assays [32]. However, its effect against *P. digitatum* had not been reported, and we found that it was also highly effective at concentrations as low as 0.62 g/L.

In the present work, SM, CN, EU, GE, VA, PRO, and MY were incorporated into PEC-based formulations for the first time in an in vivo study to evaluate their curative activity on ‘Valencia’ oranges inoculated with *P. digitatum*. As expected, the PEC coating without antifungal ingredient did not control GM after incubation at 20 °C, confirming the need for additional active antifungal ingredients in the PEC matrix. Interestingly, SM, VA, and PRO, which were very effective in the in vitro tests, did not reduce the incidence and severity of GM on coated oranges incubated at 20 °C. These differences between in vitro and in vivo results have also been reported in other works and have been attributed to the lower release rate of the active ingredient from coatings located on the rind of fruits than from artificial growing media such as PDA [11,23]. On the contrary, EU, GE, and MY showed the best results in reducing GM incidence on inoculated oranges although their effectiveness did not always increase with increasing concentrations. In general, the complex interactions between host, pathogen, and environment that occur during disease development determine the in vivo disease control ability of antifungal agents. In some cases, specific interactions between the applied active ingredient and the tissue of the fruit host may involve biochemical reactions that lead to the induction of defense mechanisms that contribute to disease control [12], whereas in other cases, they might negatively affect the inherent antifungal activity of the agent [2]. In addition, when the active agent is an ingredient of an EC, several factors, such as emulsion composition and properties, possible interactions of the antifungal compound with the coating components and volatility of the ingredient, can play an important role in the overall antifungal performance of the coating since they can affect the availability and release ability of the agent [7,8].

On inoculated ‘Valencia’ oranges coated and stored at 5 °C, the effectiveness of antifungal ECs decreased during storage, confirming that the effect of these natural agents is rather fungistatic than fungicidal. Similar findings were reported in previous studies using GRAS salts as antifungal ingredients of different composite ECs to control citrus GM [2,33]. In general, our results showed the potential of PEC-based coatings containing EU (8 g/kg) or GE (2 g/kg) to reduce GM development on ‘Valencia’ oranges both at room temperature and during long-term cold storage. To our knowledge, this is the first report on the ability of these type of composite ECs to preserve citrus fruits after harvest. Some studies have reported the effectiveness of EOs incorporated into natural (carnauba) or synthetic commercial waxes to reduce citrus molds caused by *P. digitatum* or *P. italicum*, highlighting the advantages of these materials to provide a better contact between the EO and fruit surface compared to aqueous solutions applications [15,34]. For instance, CN (5 mL/L) combined with aqueous ethanol solution (1 mL/L) reduced by 40% GM incidence on oranges artificially inoculated after 7 d at 23 °C in a preventive assay, whereas the same concentration of EO incorporated into commercial waxes achieved reductions of 62–100% depending on the type of wax [34]. In contrast, Plaza et al. [11] reported that thyme and oregano EOs (50 mL/L) added to a commercial wax coating were not effective to reduce GM incidence on ‘Valencia’ oranges incubated at room temperature. Chitosan ECs have also been studied as vehicles for the incorporation of EOs to control citrus diseases, with contradictory results. While Shao et al. [14] did not find any effect on GM control on cold-stored mandarins by clove EO added to chitosan compared to the use of chitosan alone, the application of chitosan–thyme EO reduced the incidence of blue mold caused by *P. italicum* on oranges incubated at 25 °C and was superior to chitosan alone [23]. To our knowledge, there are no available reports about the use of pectin-based ECs as carriers for natural antifungal compounds for controlling postharvest citrus diseases.

The impact of antifungal ECs on the physicochemical and sensory quality of ‘Valencia’ oranges after long-term cold storage was also evaluated in this work. In terms of weight loss, the PEC EC without EOs was not an effective barrier to water vapor, while the addition of the antifungal agents, mainly MY and EU, enhanced this function. Changes in permeability of composite coatings could be attributed to differences in the relative concentrations of the main ingredients, interactions between polymer chains and lipid components with the antifungal compounds, and a higher hydrophobicity of the coatings due to the incorporation of lipophilic compounds [34]. The reduction in weight loss by antifungal ECs did not translate into a maintenance of fruit firmness. Previous studies showed that the impact of coatings on fruit firmness is not only dependent on the coating composition but also on inherent characteristics of the citrus cultivar, being more effective in mandarins than in oranges [8,25]. In addition, the application of ECs increased the content of ethanol and acetaldehyde in the juice, showing the ability of the coatings to modify the fruit internal gas composition by creating a barrier to gases [8,25]. Minimum ethanol content associated with off-flavors in citrus fruits has been reported to be 2000 mg/L [25], whereas the maximum value observed in our work was 1149 mg/L. Accordingly, we observed in sensory tests that overall flavor and off-flavors of oranges were not negatively affected by the application of antifungal ECs. Furthermore, we found that the addition of EU to the PEC coating improved fruit gloss, probably due to differences in the optical properties of the coating according to its composition. Overall, treatments with PEC-based antifungal coatings successfully maintained the physicochemical and sensory quality of oranges throughout long-term cold storage.

## 5. Conclusions

The present study shows the potential of citrus PEC-based ECs containing EOs as a non-polluting alternative to control citrus GM and maintain the postharvest quality of ‘Valencia’ oranges. Among the different antifungal ECs evaluated, those formulated with EU or GE showed a significant curative activity against GM on artificially inoculated oranges either incubated at room temperature or stored at 5 °C. Moreover, ECs containing EU or MY satisfactorily reduced weight loss and maintained the quality of long-term cold-stored oranges, while ECs with EU improved fruit gloss. Therefore, the PEC-BW coating containing EU could be a promising commercial treatment to reduce decay and maintain postharvest quality of citrus fruits, providing a safe alternative to conventional waxes amended with synthetic chemical fungicides. Further research should focus on the evaluation of the efficacy of PEC-BW-EU coatings to control other relevant postharvest citrus diseases, such as blue mold caused by *P. italicum* and sour rot caused by *Geotrichum citri-aurantii*, and their performance when applied to other important citrus species and cultivars in order to increase their potential for commercial application.

## Figures and Tables

**Figure 1 foods-11-01083-f001:**
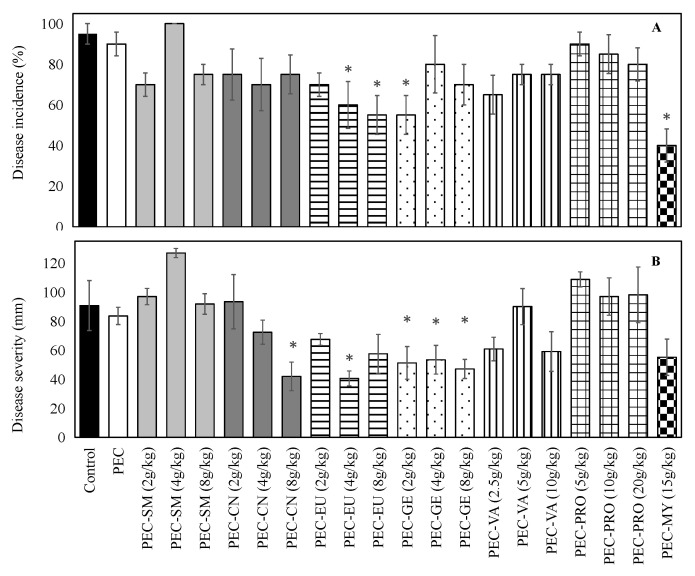
Incidence (**A**) and severity (**B**) of green mold on ‘Valencia’ oranges artificially inoculated with *Penicillium digitatum*, uncoated (control) or coated with pectin (PEC)–lipid edible composite coatings containing different concentrations of *Satureja montana* (SM), cinnamon (CN), and myrrh (MY) essential oils; eugenol (EU), geraniol (GE), vanillin (VA), or propolis extract (PRO); and incubated for 8 days at 20 °C. * indicates means significantly lower than control and PEC-coated (without antifungal agent) samples (*p* < 0.05).

**Figure 2 foods-11-01083-f002:**
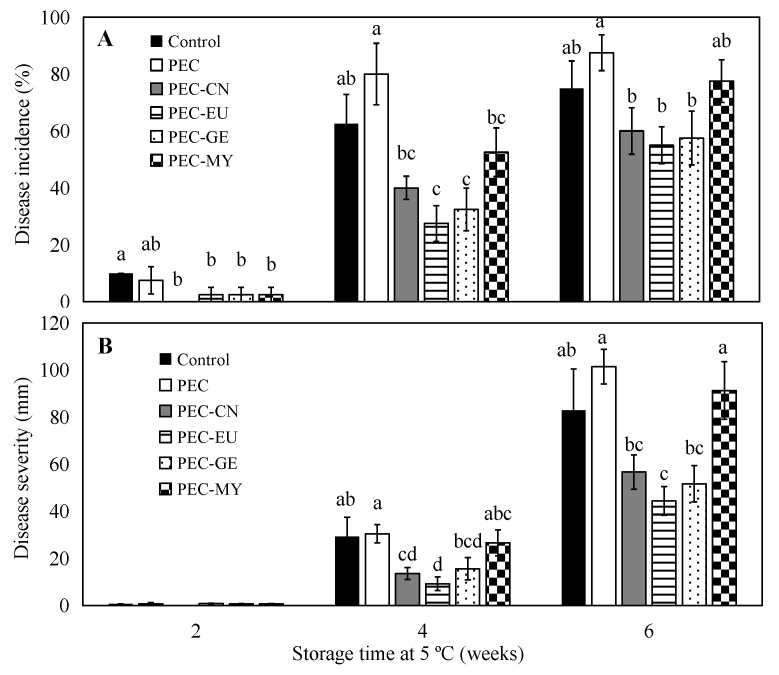
Incidence (**A**) and severity (**B**) of green mold on ‘Valencia’ oranges artificially inoculated with *Penicillium digitatum*, uncoated (control) or coated with selected pectin (PEC)–lipid edible composite coatings containing cinnamon (CN; 8 g/kg), eugenol (EU; 8 g/kg), geraniol (GE; 2 g/kg), or myrrh (MY; 15 g/kg) and stored at 5 °C and 90% RH for up to 6 weeks. For each storage time, means with different letters are significantly different according to LSD test (*p* < 0.05).

**Figure 3 foods-11-01083-f003:**
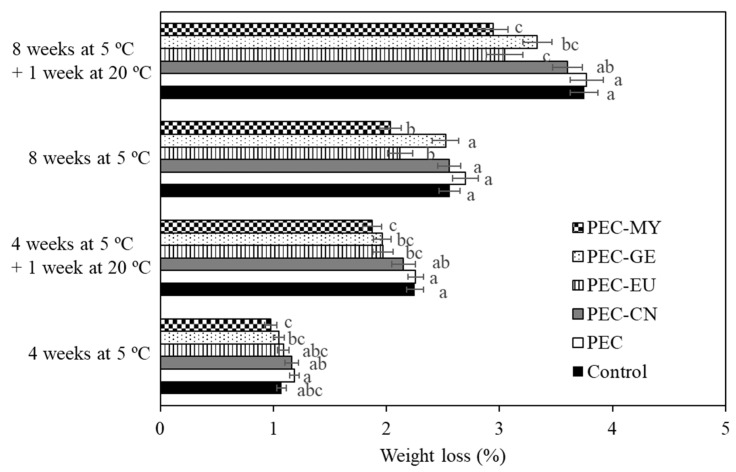
Weight loss of ‘Valencia’ oranges uncoated (control) or coated with pectin (PEC)–lipid edible coatings containing cinnamon (CN; 8 g/kg), eugenol (EU; 8 g/kg), geraniol (GE; 2 g/kg), or myrrh essential oils (MY; 15 g/kg) after different periods of cold storage and shelf life. For each storage time, means with different letters are significantly different according to LSD test (*p* < 0.05).

**Table 1 foods-11-01083-t001:** Mycelial growth inhibition of *Penicillium digitatum* exerted by essential oils (EOs) and natural extracts in in vitro tests after 7 days of incubation at 25 °C.

Volatile Exposure Method			Agar Dilution Method		
Natural Agent	Dose (µL)	Mycelial Growth Inhibition (%)	Natural Agent	Dose (g/kg)	Mycelial Growth Inhibition (%)
SM	10	98.2 ± 1.1 ^a^	GT	5.00	25.6 ± 0.4 ^c^
	20	100.0 ± 0.0 ^a^		10.00	58.1 ± 3.9 ^b^
	40	100.0 ± 0.0 ^a^		20.00	76.2 ± 1.0 ^a^
CN	10	55.7 ± 4.8 ^b^	VA	0.62	82.9 ± 3.2 ^b^
	20	98.6 ± 1.4 ^a^		1.25	100.0 ± 0.0 ^a^
	40	100.0 ± 0.0 ^a^		2.50	100.0 ± 0.0 ^a^
LG	10	0.0 ± 0.0 ^b^	MY	1.25	42.3 ± 3.6 ^a^
	20	1.9 ± 1.5 ^b^		2.50	41.6 ± 0.6 ^a^
	40	49.6 ± 5.9 ^a^		5.00	43.6 ± 0.9 ^a^
EU	10	91.2 ± 0.5 ^a^	PRO	5.00	100.0 ± 0.0 ^a^
	20	92.1 ± 2.4 ^a^		10.00	100.0 ± 0.0 ^a^
	40	94.7 ± 1.4 ^a^			
GE	10	60.3 ± 14.3 ^b^			
	20	89.1 ± 0.0 ^a^			
	40	89.9 ± 0.4 ^a^			

For each natural agent, means with different letters indicate significant differences among doses according to LSD test (*p* < 0.05). SM, *Satureja montana* EO; CN, cinnamon EO; LG, lemongrass EO; EU, eugenol; GE, geraniol; GT, green tea extract; VA, vanillin; MY, myrrh EO; PRO, propolis extract.

**Table 2 foods-11-01083-t002:** Physicochemical and sensory quality attributes of ‘Valencia’ oranges treated with antifungal pectin (PEC)-based edible coatings and stored at 5 °C followed by commercial shelf-life period of 7 days at 20 °C.

Physicochemical Quality	Storage Conditions (Weeks at 5 °C + 1 Week at 20 °C)	Treatments					
		Control	PEC	PEC-CN	PEC-EU	PEC-GE	PEC-MY
Firmness (% deformation)	At harvest	3.01 ± 0.19					
	4 weeks at 5 °C	2.88 ± 0.09 ^a^	3.01 ± 0.13 ^a^	2.99 ± 0.18 ^a^	3.08 ± 0.14 ^a^	2.95 ± 0.13 ^a^	3.05 ± 0.14 ^a^
	8 weeks at 5 °C	3.50 ± 0.20 ^ab^	4.00 ± 0.20 ^a^	2.40 ± 0.23 ^c^	3.33 ± 0.13 ^b^	3.12 ± 0.20 ^b^	3.37 ± 0.28 ^b^
Titratable acidity (g/L citric acid)	At harvest	6.54 ± 0.43					
	4 weeks at 5 °C	6.70 ± 0.55 ^a^	8.12 ± 0.35 ^a^	7.58 ± 0.31 ^a^	8.19 ± 0.46 ^a^	7.44 ± 0.23 ^a^	7.13 ± 0.59 ^a^
	8 weeks at 5 °C	7.17 ± 0.12 ^bc^	8.90 ± 0.12 ^a^	8.08 ± 0.52 ^ab^	7.93 ± 0.42 ^ab^	6.66 ± 0.26 ^c^	7.84 ± 0.31 ^bc^
Soluble solids content (°Brix)	At harvest	10.3 ± 0.1					
	4 weeks at 5 °C	12.0 ± 0.3 ^a^	11.9 ± 0.1 ^a^	10.7 ± 0.3 ^b^	10.9 ± 0.1 ^b^	11.8 ± 0.0 ^a^	11.0 ± 0.2 ^b^
	8 weeks at 5 °C	11.0 ± 0.1 ^a^	11.2 ± 0.1 ^a^	10.8 ± 0.2 ^a^	10.6 ± 0.3 ^a^	10.1 ± 0.4 ^a^	11.2 ± 0.5 ^a^
Ethanol (mg/L)	At harvest	153.6 ± 20.1					
	4 weeks at 5 °C	316.1 ± 14.9 ^d^	534.2 ± 91.5 ^bc^	810.2 ± 105.2 ^a^	712.3 ± 44.7 ^ab^	544.0 ± 51.5 ^b^	328.5 ± 57.2 ^cd^
	8 weeks at 5 °C	434.7 ± 86.9 ^d^	1039.7 ± 31.0 ^ab^	1148.9 ± 78.0 ^a^	905.9 ± 103.4 ^bc^	747.1 ± 11.6 ^c^	1011.6 ± 38.8 ^ab^
Acetaldehyde (mg/L)	At harvest	5.0 ± 0.6					
	4 weeks at 5 °C	10.5 ± 0.5 ^c^	13.2 ± 1.1 ^abc^	14.5 ± 0.9 ^a^	15.0 ± 0.8 ^a^	13.8 ± 0.7 ^ab^	11.7 ± 1.2 ^bc^
	8 weeks at 5 °C	10.5 ± 1.4 ^b^	18.4 ± 0.3 ^a^	20.5 ± 0.5 ^a^	18.6 ± 1.5 ^a^	19.1 ± 0.4 ^a^	19.3 ± 0.5 ^a^
**Sensory Quality**							
Overall flavor	At harvest	7.3 ± 0.3					
	4 weeks at 5 °C	6.8 ± 0.4 ^ns^	6.6 ± 0.5	6.9 ± 0.5	6.8 ± 0.3	7.0 ± 0.7	7.1 ± 0.4
	8 weeks at 5 °C	4.4 ± 0.5 ^ns^	4.9 ± 0.2	5.4 ± 0.4	4.6 ± 0.4	4.0 ± 0.7	5.8 ± 0.4
Off-flavor	At harvest	1.1 ± 0.1					
	4 weeks at 5 °C	1.2 ± 0.1 ^ns^	1.1 ± 0.1	1.1 ± 0.1	1.1 ± 0.1	1.1 ± 0.1	1.0 ± 0.0
	8 weeks at 5 °C	1.8 ± 0.4 ^ns^	1.8 ± 0.4	1.5 ± 0.2	1.6 ± 0.3	2.0 ± 0.5	1.4 ± 0.3
Visual quality	At harvest	3.0 ± 0.0					
	4 weeks at 5 °C	3.0 ± 0.0 ^ns^	3.0 ± 0.0	3.0 ± 0.0	3.0 ± 0.0	3.0 ± 0.0	3.0 ± 0.0
	8 weeks at 5 °C	2.4 ± 0.2 ^ns^	2.5 ± 0.3	2.6 ± 0.2	2.5 ± 0.3	2.4 ± 0.3	2.4 ± 0.3

For each quality attribute and storage period, different letters and “ns” in each row indicate significant and non-significant differences among treatments, respectively, according to LSD test (*p* < 0.05). CN, cinnamon EO (8 g/kg); EU, eugenol (8 g/kg); GE, geraniol (2 g/kg); MY, myrrh EO (15 g/kg). Overall flavor rated from 1 (very poor) to 9 (optimum), off-flavor from 1 (absence) to 5 (high presence), and visual quality from 1 (bad) to 3 (good).

**Table 3 foods-11-01083-t003:** Ranked gloss of ‘Valencia’ oranges coated with antifungal pectin (PEC)-based edible coatings and stored at 5 °C followed by 1 week of shelf life at 20 °C.

Gloss Rank	Storage Conditions	
	4 Weeks at 5 °C + 1 Week at 20 °C	8 Weeks at 5 °C + 1 Week at 20 °C
Glossier	PEC-EU ^a^	PEC-CN ^ns^
	PEC-GE ^ab^	PEC-EU
	PEC-CN ^abc^	Control
	PEC-MY ^bc^	PEC-GE
	Control ^bc^	PEC-MY
Less glossy	PEC ^c^	PEC

Treatments in columns with different letters are significantly different according to Friedman test (*p* < 0.05) (*n* = 10). ns, non-significant differences; CN, cinnamon essential oil (8 g/kg); EU, eugenol (8 g/kg); GE, geraniol (2 g/kg); MY, myrrh essential oil (15 g/kg).

## Data Availability

The data presented in this study belong to the IVIA and are available on request from the corresponding author.

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
