# Peer review of "Natural Pectin-Based Edible Composite Coatings with Antifungal Properties to Control Green Mold and Reduce Losses of ‘Valencia’ Oranges"

_foods, 2022, doi:10.3390/foods11081083_

Round 1

Reviewer 1 Report

The manuscript is written with clear understanding of the project addressed. However, there are major concerns that need to be addressed to enhance the quality of the manuscript. My specific comments are as follows:

Abstract:

Add the methodology part before explaining the results.

Introduction:

P2L46: “In this sense, updated regulations from…” what kind of regulations. Please elaborate

P2L54: Add reference

P2L72: “Several studies have reported the antifungal potential of…” Discuss the findings

P2L82: “In this work, several natural plant extracts…” list down the extracts used in this study

Materials and Methods:

Section 2.5 should be in the first section (2.1) since it indicates the sample preparation of this study

P5L182: “Lots of 70 oranges per treatment were manually coated..” explain briefly how 70 oranges were used

Section 2.7.2: Why the number of oranges per treatment is not similar for each quality attributes? For examples, 20 oranges were used for fruit weight loss. 10 oranges for firmness? What’s the actual reasoning for the difference number of samples per quality attributes?

P5L199: “Volatile compounds content was determined from the headspace of 5-mL juice samples by gas chromatography.” Elaborate more on the methods so that the readers can get the clear picture of the process

Results:

P5L221: “MY EO showed no activity when tested in the vapor phase (data not shown), so it was evaluated by the agar dilution method” contradict with next statement “MY EO caused a moderate effect that did not depend on the concentration” Please justify

Please improve the quality of Figure 1, especially the x-axis

P6L249: “After 2 weeks of cold storage, all the antifungal ECs reduced disease incidence compared to control samples” contradict with the figure. After 2 weeks, PEC increased

P6L252: “At the end of the storage period, disease incidence increased for all treatments, without significant differences between control and coated oranges.” Revise this sentence, it is very confusing.

Please improve the quality and the interpretation of Figure 2. For Figure 2(b), put the legend

P7L274: The explanation in the text for Figure 3 seems like completely different from the figure since it mentions 7 days of shelf life at 20 C.

Figure 3. I don’t understand the caption for the y-axis. How about the results for 5 C? please improve the quality. caption on the x-axis is not clear

P8L286: “…followed by a shelf-life period of 1 week at 20 °C..” the readers might be confused whether this is another treatment or not. Please revise

Comparison of the mean values, together with the standard errors, obtained for physicochemical quality after different storage conditions shows obvious errors in the presented results of the statistical analysis. The letters assigned to each homogeneous group should be ordered ascending or descending relative to the mean value. The results of the statistical analysis presented in Table 2 are neither of these cases. The order of the homogeneous groups is further confused.

Discussions:

P10L338: “some of the key active compounds of Eos..” what are key active compounds?

P10L346: “This dose was higher than that reported by..” I understand that the dose for the current study is higher than reported in literatures. I think it would be good if the authors try other EU doses lower than 5µL.

P10L360: “…‘volatile exposure’ method.” What does it mean by volatile exposure method?

P11L389: “As expected, the PEC coating without antifungal ingredient did not control GM after incubation at 20 °C…” did not control in terms of what?

P11L409: “On inoculated ‘Valencia’ oranges coated and stored at 5 °C, the effectiveness of anti-fungal ECs decreased during storage…”how about the samples stored at 20 C?

is it really make a difference if it stored followed by 1 week at 20 °C? what is the justification of doing this?

Conclusion:  

Author need to improve more in the conclusions section. The conclusion is vague and too general. This section did not conclude the objective of the paper. Add your own point of view to provide the overall critical finding. How about the impact of these treatments on postharvest quality at 20 °C?

General: Please check the English language. Also, for spelling and grammatical errors.

Reviewer 2 Report

Comments to authors

Journal: Foods

Title: Natural Pectin-Based Edible Composite Coatings with Antifungal Properties to Control Green Mold and Reduce Losses of ‘Valencia’ Oranges

The current article focused on the formation of Pectin-based edible coating with the addition of natural extracts and edible oils to improve the anti-fungal properties of the oranges. This article is

interesting, well documented and under the domain of the journal but still, I have fewer concerns as mentioned below.

  1. The introduction of the manuscript should be improved and well-structured there are few grammatical errors throughout the manuscript.
  2. Why do authors say this material is Novel? Can you please explain more?
  3. Which method did the authors use to find out Mycelial growth inhibition (%)? How did they know the significance level by using different concentrations of each natural agent?
  4. What technique and formula were used to measure the disease incidence and disease severity? Authors should also mention in the manuscript’s material and method part.
  5. Antioxidant activity and phenolic profile determination through HPLC analysis.
  6. As volatile compound contents are very important for this study, I suggest authors also mention gas chromatography results in this manuscript for more authenticity.
  7. The authors should also provide the antimicrobial properties of the final product.
  8. The nutritional properties of the fruit will also be affected by coating. Why authors did not mention the nutritional profile of the coated product?
  9. Results should be well defined and the discussion part is not closely related to the results, and the focus of the discussion is not focused. So, improve the discussion part according to the current and previous studies.

Reviewer 3 Report

Review on manuscript: foods-1650793

Natural Pectin-Based Edible Composite Coatings with Antifungal Properties to Control Green Mold and Reduce Losses of Valencia Oranges

by María Victoria Alvarez, Lluís Palou, Verònica Taberner, Asunción Fernández-Catalán, Maricruz Argente-Sanchis, Eleni Pitta, María Bernardita Pérez-Gago

submitted to Foods

In the manuscript submitted for evaluation, the authors characterized pectin-based edible coatings with antifungal properties to reduce green mold of oranges.

The topic undertaken by the authors is current and important from the point of view of food quality and safety. The manuscript is generally prepared correctly and needs some corrections.

Detailed recommendation:

line 99 – does propolis mean its ethanolic extract?

line 147 – instead of cP, the SI unit should be used,

lines 192-193 – more information should be added, dimensions of the measuring element, travel speed, operating time, etc.,

line 200 – more information about GC analyses should be added, which specific compounds were determined?

line 203 – full scale should be presented,

Table 1 – is the marking of significance of differences between means correct? does 89.1 ± 0.0 not differ significantly from 100.0 ± 0.0?

Figure 2 – quality should be improved,

lines 328-334 –  this fragment of the text adds nothing important.

Round 2

Reviewer 2 Report

Authors successfully defended the question and improve the required part of the manuscript as per suggestions. The manuscript has been greatly improved.